# Impact of Rice Bran Oil Emulsified Formulation on Digestion and Glycemic Response to Japonica Rice: An In Vitro Test and a Clinical Trial in Adult Men

**DOI:** 10.3390/foods13162628

**Published:** 2024-08-21

**Authors:** Naoki Kawada, Keiko Kamachi, Masatsugu Tamura, Maki Tamura, Rika Kitamura, Kenta Susaki, Hiroyoshi Yamamoto, Hideaki Kobayashi, Ryosuke Matsuoka, Osamu Ishihara

**Affiliations:** 1R&D Division, Kewpie Corporation, 2-5-7 Sengawa Kewport, Sengawa-Cho, Chofu-shi 182-0002, Tokyo, Japan; 2Nutrition Clinic, Kagawa Nutrition University, Komagome, Toshima-ku 170-8481, Tokyo, Japan; 3School of Agriculture, Utsunomiya University, 350, Mine-machi, Utsunomiya-shi 321-8505, Tochigi, Japan

**Keywords:** blood glucose, rice, rice bran oil, emulsified formulation, continuous glucose monitoring, clinical trial, in vitro starch digestion

## Abstract

To assess the effect of rice bran oil emulsified formulation (EMF) on cooked rice, a single-arm open clinical trial and in vitro testing for digestion and glycemic response were performed. Fifteen Japanese men consumed 200 g of packed rice, cooked with or without EMF. Blood samples were collected 0, 30, 60, and 120 min post-consumption and analyzed for glucose, insulin, and triglyceride levels. Continuous glucose monitoring (CGM) and sensory evaluation were also performed. A two-step in vitro digestion test, simulating gastric and small intestinal digestion was conducted. EMF-added rice group showed higher insulin response levels at 60 min than the placebo group. Stratification of participants with HbA1c ≥ 5.6 or an insulinogenic index ≤ 0.4 revealed a significant reduction in C_max_ glucose levels. A significant correlation was observed between venous and CGM blood glucose levels and no significant sensory differences were observed. The in vitro test revealed significantly lower C_∞_, equilibrium starch concentrations, with EMF. Clinical trial suggests that EMF may stimulate insulin secretion and reduce blood glucose levels in participants with lower insulin responses. In vitro tests suggest that EMF inhibits glycemic digestion. This trial was registered at the UMIN Center (UMIN000053495; registered 31 January 2024).

## 1. Introduction

Diabetes mellitus is a group of carbohydrate metabolism disorders diagnosed by glucose and HbA1c levels. Participants may be classified by type1 diabetes (T1D) or type2 diabetes (T2D), of which the latter can potentially be controlled through dietary modification [1]. A general consensus has not reached an ideal dietary balance for people with glucose intolerance [2,3]; however, some dietary patterns, foods, and nutrients, such as Mediterranean food, low glycemic index (GI) food, and dietary fiber reduce the risk of T2D [4,5,6,7].

White rice is a staple food in many Asian countries; however, there exist reports that its consumption is associated with a higher risk of T2D due to the rapid increase in blood glucose [8]. Packed rice, whose consumption has recently increased with several social changes, such as the increase in single-person households and the social advancement of women, is also potentially related to a high GI [9,10,11]. Therefore, an emerging need for a healthy and more palatable way of eating white rice and packed rice is attracting.

Lipids and emulsified lipids are widely used in food processing to modify the features of starch, such as improvement of paste properties and resistance to retrogradation by forming amylose-lipid complexes [12,13,14]. Adding lipids to cooked rice reportedly changes the characteristics of starch [15,16] and decreases in vitro starch digestibility [17,18,19,20]. In a clinical trial, coconut oil was shown to modulate postprandial blood glucose levels in obese children [21]. Furthermore, canola oil emulsified formulation (EMF) modulated the increase in postprandial blood glucose levels in participants with high body mass index (BMI) [22]. However, few clinical studies have reported the effects of adding lipids on the glycemic response and the relationship between clinical data and in vitro digestibility tests. For example, a regulatory effect of rice bran oil has been reported in in vitro starch digestibility tests but not in clinical trials [17,18].

In this study, the glycemic digestibility and response of packed rice with a small amount of rice bran oil EMF were assessed through both in vitro tests and clinical trials to evaluate its clinical effect and to clarify the underlying mechanism. To provide stability and uniformity of test foods, rice bran oil was added as an EMF. In the clinical trial, continuous glucose monitoring (CGM) was used alongside venous blood sampling analysis. In addition, the taste of the rice was assessed through a self-administered questionnaire.

## 2. Materials and Methods

### 2.1. Test Foods

Two types of packed rice, including 200 g of standard rice and special rice cooked with a small amount of rice bran oil EMF, were provided by TableMark Co., Ltd. (Tokyo, Japan). Japonica rice, *Oryza sativa* L. cv. Koshihikari, produced in Niigata, Japan, in 2023 was used as test food. Regarding the amount of EMF, the weight ratio of lipids to cooked rice was 0.6%. To facilitate industrial application, this amount was determined by the quantity of oil typically used in commercial rice balls and packed rice. The manufacturing process of packed rice was as follows: (1) polishing, (2) washing and soaking, (3) filling in the cup container, (4) high temperature and pressure short-time sterilization, (5) cooking with/without EMF, (6) sealing and packing, and (7) cooling [9]. The EMF used in this study was improved from a previously described formulation [22], and the materials are presented in Table 1. The nutrient content of each test food was analyzed by Japan Food Research Laboratories (Tokyo, Japan) and is listed in Table 2.

### 2.2. Participants

Twenty-nine Japanese male participated in the initial screening test using standard rice. This test excluded several subjects, i.e., one with diabetic type, three who presented multiple hypoglycemic events as indicated by CGM a few days following the initial test, and 10 participants with possible abnormalities in postprandial blood glucose (their CGM blood glucose levels did not match their daily dietary records). Consequently, fifteen subjects were enrolled based on assessments by a medical doctor and a registered dietitian.

The sample size of participants was determined based on a previous clinical trial on the effects of cooked rice on postprandial blood glucose [23]. Using the Cancer Research and Biostatistics Statistical Tools “https://stattools.crab.org/ (accessed on 13 August 2024)”, a sample size of 10 participants per group was required with a power (1-β) of 80% and an alpha (α) of 0.05.

### 2.3. Study Design

This clinical trial was a single-arm open study conducted in February 2024. The study flow diagram is shown in Figure 1. 29 participants ingested 200 g of normal packed rice as the placebo test food. Fifteen subjects selected by the above reasons ingested the packed rice with rice bran oil EMF as the treatment test food. They fasted since 9 p.m. on the previous day of their visit. Participants remained in a seated position for two hours after eating and blood samples were collected before and 30, 60, and 120 min after food consumption. Both packed rice formulations were heated for 10 min at 100 °C in a steam convection oven (iCombi Classic, Rational Japan Co., Ltd., Tokyo, Japan) and served to the participants, who ingested the food within 8 min and chewed 15 times per bite.

### 2.4. Measurements and Analysis

Blood glucose, insulin, and triglyceride (TG) levels were measured at each injection point. In addition, anthropometric measurements, physical examinations, hematological tests, blood biochemistry tests, and medical interviews were conducted before ingestion. The blood analysis was done at the LSI Medience Corporation (Tokyo, Japan).

### 2.5. Sensory Evaluation

Self-administered questionnaires for the sensory evaluation were completed by the participants. The questionnaires consisted of six items: appearance, flavor, taste, adhesiveness, hardness, and overall evaluation, and the participants answered based on a seven-point scale (−3, −2, −1, 0, 1, 2, and 3) [24].

### 2.6. Continuous Glucose Monitor

29 participants wore a CGM device, a Dexcom G6 CGM (Dexcom, Inc., San Diego, CA, USA), at the placebo test before ingestion. This device can measure glucose concentration in the interstitial fluid every 5 min. Participants wore it for more than 2 days and returned it to the clinic. The fifteen subjects attached the CGM device again before eating the EMF test food and removed it 180 min after ingestion. To prevent a change in dietary habits, the monitor screen of the CGM device was set so that the participants were blinded.

### 2.7. In Vitro Digestibility Test

#### 2.7.1. Simulated Digestive Fluid Preparation

Pepsin (P7000, porcine gastric mucosa, ≥250 U/mg, solid), pancreatin (hog pancreas, 4 × USP), and invertase (baker’s yeast, grade VII, ≥300 U/mg, solid) were purchased from Sigma-Aldrich (Waltham, MA, USA). Amyloglucosidase (3260 U/mL) was purchased from Megazyme (Megazyme International, Wicklow, Ireland). Simulated gastric fluid, pepsin dissolved in sodium chloride solution (pH 1.20), and simulated intestinal fluid, pancreatin, invertase, and amyloglucosidase dissolved in potassium dihydrogen phosphate solution (pH 6.80), were prepared based on previous reports [25,26,27].

#### 2.7.2. In Vitro Digestibility Test Protocol

The packed rice was heated in a microwave oven at 500 W for 2 min and left at room temperature for 30 min. A two-step in vitro digestion test simulating gastric and small intestinal digestion was done to evaluate the digestibility of the test foods [27,28]. A sample equivalent to 6.8 g of total starch in a polyethylene net with adding distilled water to a total of 170 g was suspended in a double-tube reactor at 37 °C. The sample in the polyethylene net was stirred without contacting to the inside of the case at 300 rpm (color squid, Ika). The pH was constantly measured (AS800, As One) and maintained at 1.20 ± 0.02 for the gastric digestion phase and 6.80 ± 0.02 for the small intestinal digestion phase by adding hydrochloric acid and sodium hydroxide. At 5 and 30 min of gastric digestion and 5, 10, 20, 30, 60, 120, 180, 240, 300, and 360 min of small intestinal digestion, 0.5 mL of the supernatant was collected and 3 mL of 95% ethanol were added to stop the enzyme reaction. The amount of glucose was measured by a D-Glucose Assay Kit (GOPOD Format K-GLUK 08/23, Megazyme International, Wicklow, Ireland) after treatment with an amyloglucosidase-invertase mixture. The carbohydrate digestibility (%) was calculated as the starch hydrolysis rate based on the glucose content using the following equation:(1)$$S_{H}=\frac{S_{h}}{S_{i}}=0.9\frac{G_{p}}{S_{i}},$$
where *S_H_* is the carbohydrate digestibility (%), *S_h_* is the amount of hydrolyzed starch, *S_i_* is the initial amount of starch, and *G_p_* is the amount of glucose produced. The conversion factor (0.9) for the molecular weight of the glucose units in the starch/molecular weight of glucose (162/180 = 0.9). A first-order equation established by Goñi et al. [28] was used to describe the kinetics of starch hydrolysis during intestinal digestion and the parameters were estimated using Igor Pro software (version 4.01; Hulinks, Tokyo, Japan) as follows:(2)$$C_{t}=C_{\infty }\;\left(1-e^{-kt}\right),$$
where *C_t_* is the percentage of hydrolyzed starch at time t, *C*_∞_ is the equilibrium concentration of starch, and *k* is the kinetic constant. The estimated glycemic index (eGI) was calculated by fitting the equation proposed [28] and using a crumb of bread (Choujuku; Pasco Shikishima Corp., Aichi, Japan) as the standard sample.

### 2.8. Statistical Analysis

The results are presented as the mean ± standard deviation (SD). Wilcoxon’s signed-rank test was performed to compare clinical trial data and sensory evaluation data between groups. Stratified analyses for the clinical trial were conducted in subjects with an HbA1c ≥ 5.6 and insulinogenic index (IGI) ≤ 0.4. IGI was used as an index of β-cell function and was calculated as follows [29]:(3)$$Insulinogenic\;index=\frac{\Delta Insulin\left(30\;\min-0\;\min\right)}{\Delta Glucose\left(30\;\min-0\;\min\right)},$$

Welch’s two-sample *t*-test was performed to compare the in vitro test data. The correlations between venous blood glucose and CGM blood glucose were assessed by Pearson correlation test. Statistical significance was indicated at *p* < 0.05. IBM SPSS Statistics 28.0 (IBM Japan Co., Ltd., Tokyo, Japan) was used to perform the statistical analyses.

## 3. Results

### 3.1. Analysis of the Blood Samples

The characteristics of all subjects are listed in Table 3. The results of the blood measurements are shown in Figure 2. Blood insulin levels at 60 min after intake of the EMF test food were higher compared with that of the placebo test food. No significant differences were observed for the other items.

### 3.2. Stratified Analysis of the Blood Samples

To assess the potential benefit of EMF rice, stratified analyses were conducted for the participants considered to be at high risk for diabetes.

Firstly, nine subjects with an HbA1c ≥ 5.6, considered to be at high risk by the Japanese Clinical Practice Guideline for Diabetes, were examined [30]. The characteristics of the individuals are listed in Table 4 and the results of the blood measurements are shown in Figure 3. The C_max_ for the blood glucose levels after consuming the EMF test food was significantly lower compared with that of the placebo test food. No significant differences were observed for the other items.

Secondly, subjects with a lower IGI were evaluated. The committee of the Japan diabetes society reported that people with 0.4 μU/mL per mg/dL or lower IGI are at high risk of progression to diabetes mellitus [31]. IGI is usually used when conducting the 75 g oral glucose tolerance test, but it has been applied to other foods as well [32,33]. For this test, 200 g of normal packed white rice with 68 g of carbohydrates was considered a standard diet. Eleven participants with an IGI ≤ 0.4 μU/mL per mg/dL were evaluated. The characteristics of the individuals with HbA1c ≥ 5.6 or IGI ≤ 0.4 are listed in Table 4 and the results of the blood measurements are shown in Figure 4. After consuming the EMF test food, blood glucose levels at 60 min and the C_max_ of the blood glucose levels were significantly lower. Blood insulin levels at 30 and 60 min were higher compared with placebo test food. No significant differences were observed for the other items. Because of the addition of lipids to the rice, TG levels were also assessed at each time point (Figure 5). There were no significant differences between both test foods for all participants (*N* = 15), participants with high HbA1c (*N* = 9), and subjects with a low IGI (*N* = 11).

### 3.3. CGM Data

CGM data were partly missing for some subjects for approximately 40 min following food consumption due to communication errors in visit 1. The results included a total of 142 points (20 points before intake, 34 points at 30 min, 44 points at 60 min, and 44 points at 120 min). Subjects who participated in the only placebo test were also pooled and a correlation analysis between venous blood glucose and CGM blood glucose was performed. A significant correlation (Pearson’s correlation r > 0.7; *p* < 0.01) was observed with a linear approximation R^2^ value of 0.57. CGM blood glucose levels after consuming the EMF test food were significantly higher at 85–95 min post-intake and lower at 145–160 min post-intake compared with after consuming the placebo test food (Appendix A).

### 3.4. In Vitro Digestibility Test

Figure 6 shows the changes in the starch digestibility rate for the placebo and EMF test foods. Gastric and small-intestinal digestion were simulated during the initial 30 min and the following 360 min. The starch hydrolysis rate in the gastric digestion phase ranged from 0.5 to 1.3% because of the lack of starch-digesting enzymes in the stomach. The starch hydrolysis rate in the small intestinal digestion phase increased rapidly, whereas that of EMF test food was lower compared with that of placebo test food after 150 min. The kinetic parameters of starch hydrolysis are listed in Table 5. C_∞_ of the EMF test food was significantly lower compared with that of the placebo test food and no significant differences were confirmed in k and eGI.

### 3.5. Sensory Evaluation

Figure 7 shows the average score of the self-administered questionnaires for sensory evaluation. No significant differences were observed for all categories between the placebo and EMF test foods.

## 4. Discussion

In this study, the glycemic digestibility and response of packed rice with a small amount of rice bran oil EMF were examined through in vitro tests and a clinical trial. This quantity of added EMF was half of that described previously [22].

In all 15 participants, insulin levels at 60 min after ingesting the EMF test food were significantly higher compared with that after ingesting the placebo test food. This may be attributed to the 1.2 g of added oils as the EMF (Table 2), which may stimulate insulin secretion. Oral ingestion of lipid emulsion reportedly elicited an insulin response by stimulating incretin secretion [34]. No significant differences were observed for the other items; however, stratified analyses showed the potential effect on the glycemic response.

Nine subjects with an HbA1c ≥ 5.6 showed significantly lower C_max_ blood glucose after ingesting EMF test food compared to placebo test food. In addition, eleven subjects with an IGI ≦ 0.4 presented significantly lower blood glucose levels at 60 min and C_max_ blood glucose, as well as significantly higher blood insulin levels at 30 and 60 min after eating EMF test foods compared to the placebo test food. Therefore, rice bran oil EMF potentially decrease blood glucose levels in subjects with poor blood sugar control and low insulin secretion [30,31] by promoting insulin secretion. However, for individuals with higher HbA1c, there are no significant differences in insulin levels. The reason of this disagreement can be explained by that they have also insulin resistance. Their higher concentrations of insulin to maintain normal insulin function [35] may masked the effects of EMF. On the other hand, EMF has also potential to inhibit or delay glucose absorption. In vitro testing showed the inhibition of glycemic absorption. Adding plant oil to starch, including rice starch, created an amylose-lipid complex and increased the content of resistant starch (RS) [12]. This complex is known as resistant starch type 5 (RS5) and occurs naturally in starch [36,37]. This EMF test food may have contained RS5 and decreased C_max_. As the effect of brown rice, including rice bran and germ, on blood glucose-related markers is well-known [38,39], the trace ingredients such as γ-oryzanol, phytosterol, tocols in the rice bran oil EMF may contribute to this effect [40]. These results suggest that EMF modulates the increase of postprandial blood glucose levels in subjects with an HbA1c ≧ 5.6 or participants with an IGI ≦ 0.4, who could be at high risk for diabetes mellitus.

The previous report using canola oil EMF showed a similar trend. Although it reported a potential effect in a stratified analysis of subjects with a BMI greater than 22 [22], the subjects showed a higher HbA1c. Additionally, the association between BMI and HbA1c both in healthy [41] and in diabetic [42,43] subjects was reported.

A significant correlation was observed between venous and CGM blood glucose levels, with a Pearson’s correlation coefficient of 0.756 and an R-squared value of 0.571 for a linear approximation. The intercept of the equation was 49 and CGM blood glucose levels tended to be above that of the venous blood glucose levels. This subtle inconsistency could be caused by the lag in CGM values compared to venous blood glucose levels [44] and the lower accuracy on the first day of attachment [45]. CGM blood glucose levels at 85, 90, and 95 min after ingestion of the EMF test food were significantly lower, and at 145, 150, 155, and 160 min, they were higher compared with the placebo test food (Appendix A). These differences may be attributed to insulin secretion at 60 min, inhibition of digestion and absorption, and RS generated by oils and fats. Although the CGM values before ingestion between the two types of test foods could not be confirmed as similar, venous blood glucose levels were confirmed.

Based on an in vitro test, the C_∞_ of the EMF test food was significantly lower compared with that of the placebo test food. The difference in starch hydrolysis after 180 min, which is the middle to late small intestinal digestion phase, is thought to be attributed to the formation of RS5 and indigestibility of oil and fat adhered to the surface of the rice grain. These data may support the gently falling postprandial CGM blood glucose levels of the EMF test food. The effect on rice grains with a 0.6% addition of rice bran oil EMF was observed in this study; however, the previous study reported no effect on rice grain digestibility with a 1.2% addition of EMF [46]. While the positive effects on rice starch with an addition exceeding 0.2% EMF [47] and on rice grains with an addition over 0.4% EMF have been reported [18]. These variations in the effects of EMF may be attributed to the rice varieties or the types of fats used in the EMF. Therefore, it is possible to significantly reduce the C_∞_, which describes a type of carbohydrate digestibility when adding at least 0.6% rice bran oil EMF to the rice grains of *Oryza sativa* L. cv. Koshihikari.

Regarding sensory evaluation, significant differences between the two test foods were not apparent. Adding 0.6% rice bran oil EMF did not spoil the flavor of the white rice. In this test, the packed rice immediately after heating was used; however, a positive effect may be achieved when using commercially distributed room temperature or cooled rice because an emulsifier is known to prevent retrogradation and extend the shelf-life [48,49,50].

This study had some limitations. The effect of a small amount of EMF on blood glucose was still unclear because no significant differences in blood glucose levels between the two types of test foods were evident in all subjects, instead of some significant differences occurring in the stratified analysis. This study included only 15 adult male subjects and a larger scale study including women and younger people should be done in the future. As mentioned above, there is a lack of CGM data; thus, the accuracy of the CGM blood glucose levels should be interpreted carefully.

## 5. Conclusions

A single-arm open clinical trial and in vitro test were performed to assess the effect of rice bran oil EMF on japonica rice in terms of digestion and glycemic response. In the clinical trial, higher insulin level at 60 min post-EMF ingestion compared with the placebo was observed. Stratified analyses of nine participants with an HbA1c ≥ 5.6 and 11 subjects with an IG index ≤ 0.4 revealed a significant reduction in C_max_ for glucose levels. The in vitro test revealed a significantly lower C_∞_, equilibrium starch concentrations, in the EMF rice. These results suggested that adding rice bran oil EMF may ameliorate blood glucose levels for subjects with low insulin secretion by stimulating insulin and delaying starch digestion. Additionally, no significant sensory differences were observed. Therefore, rice bran oil EMF may contribute to a healthy, tasty and convenient way of eating cooked white rice and packed white rice in the future.

## Figures and Tables

**Figure 1 foods-13-02628-f001:**
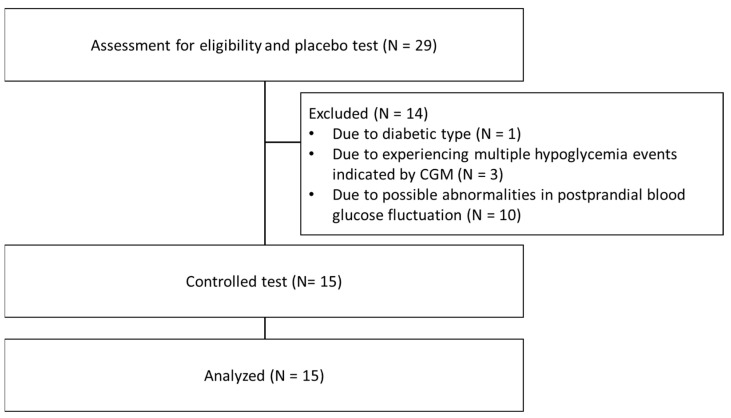
Flow chart of patients included in the study.

**Figure 2 foods-13-02628-f002:**
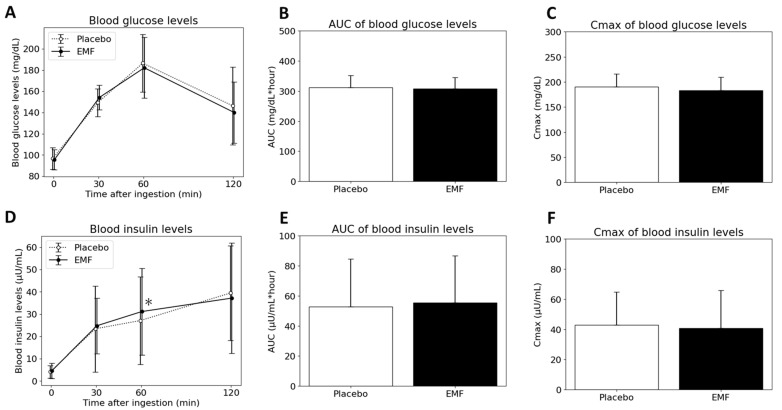
Time course of blood glucose (**A**) and insulin levels (**D**) at 0, 30, 60, and 120 min after ingestion, area under curve (AUC) of blood glucose (**B**) and insulin levels (**E**), and Cmax of blood glucose (**C**) and insulin levels (**F**). Data are presented as mean ± SD. * Significant difference between the two types of test foods.

**Figure 3 foods-13-02628-f003:**
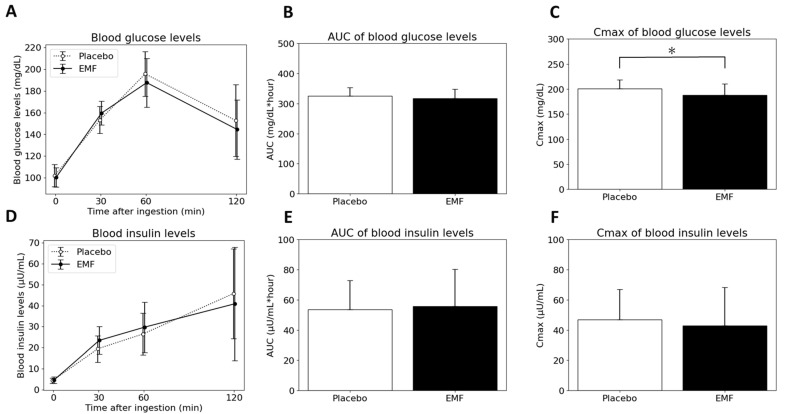
Time course of blood glucose (**A**) and insulin levels (**D**) 0, 30, 60 and 120 min after ingestion, area under curve (AUC) of blood glucose (**B**) and insulin levels (**E**), and Cmax of blood glucose (**C**) and insulin levels (**F**) for nine participants with HbA1c ≥ 5.6. Data are presented as mean ± SD. * Significant difference between the two types of test foods.

**Figure 4 foods-13-02628-f004:**
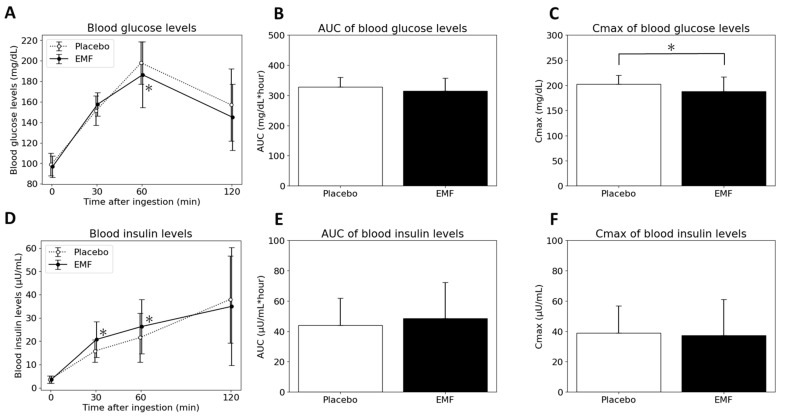
Time course of blood glucose (**A**) and insulin levels (**D**) 0, 30, 60 and 120 min after ingestion, area under curve (AUC) of blood glucose (**B**) and insulin levels (**E**), and Cmax of blood glucose (**C**) and insulin levels (**F**) for In eleven participants with IGI ≤ 0.4. Data are presented as mean ± SD. * Significant difference between the two types of test foods.

**Figure 5 foods-13-02628-f005:**
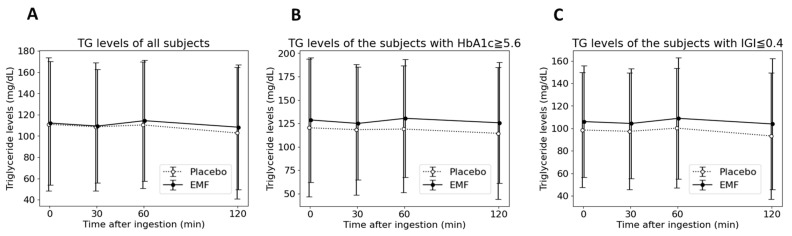
Time course of triglyceride levels at 0, 30, 60 120 min after ingestion for all participants (**A**), for nine participants with HbA1c ≥ 5.6 (**B**), and for eleven participants with IGI ≤ 0.4 (**C**). Data are presented as mean ± SD.

**Figure 6 foods-13-02628-f006:**
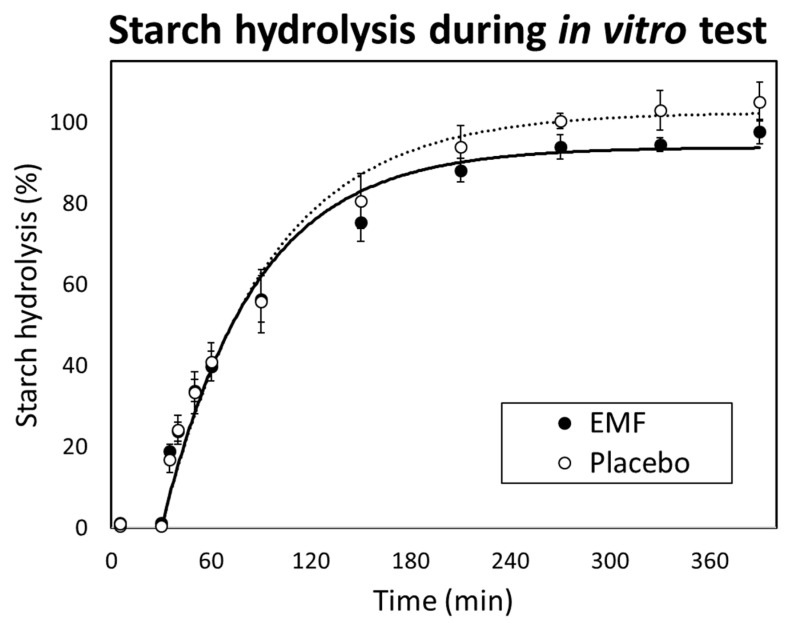
Changes in starch hydrolysis (%) of two types of packed rice during in vitro starch digestion. Error bars represent standard deviations (*n* = 4). Time from 0–30 min simulated the gastric digestion stage and 30–390 min simulated the small intestinal digestion stage.

**Figure 7 foods-13-02628-f007:**
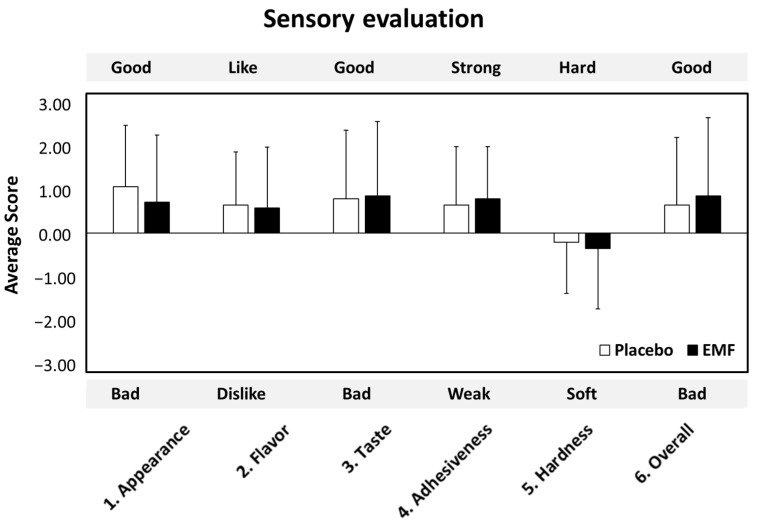
Sensory evaluations. Data are presented as mean ± SD.

**Table 1 foods-13-02628-t001:** Materials for emulsified formation (EMF).

Materials	Weight (%)
Refined rice bran oil	55.5
Reduced sugar syrup	25.0
Water	14.7
Salt	3.00
Others	1.82
Total	100

**Table 2 foods-13-02628-t002:** Nutrient content of the test food (200 g).

	Placebo	Treatment
Energy (kcal)	296	306
Protein (g)	4.00	3.60
Fat (g)	0.80	2.00
Carbohydrate (g)	68.4	68.6
Salt equivalent (g)	0.01	0.07
Water (g)	127	126

**Table 3 foods-13-02628-t003:** Clinical characteristics of participants in the clinical trial.

	Participants (*N* = 15)
Age (years)	59.4 ± 7.7
Height (cm)	172 ± 6
Body weight (kg)	74.1 ± 8.5
BMI (kg/cm^2^)	25.0 ± 2.3
Body fat percentage (%)	24.2 ± 3.9
Systolic blood pressure (mmHg)	126 ± 11
Diastolic blood pressure (mmHg)	84.1 ± 10.0
Fasting blood glucose (mg/dL)	96.8 ± 10.2
Fasting blood insulin (μU/mL)	4.17 ± 2.80
HbA1c (%)	5.55 ± 0.31
Triglyceride (mg/dL)	111 ± 63
Total cholesterol (mg/dL)	219 ± 37
LDL-cholesterol (mg/dL)	133 ± 35

All data are presented as mean ± SD. BMI, body mass index.

**Table 4 foods-13-02628-t004:** Clinical characteristics of the participants in the stratified analysis.

	Participants with HbA1c Greater than 5.6 (*N* = 9)	Participants with an Insulinogenic Index Lower than 0.4 (*N* = 11)
Age (years)	57.4 ± 7.9	58.2 ± 7.7
Height (cm)	174 ± 4	173 ± 6
Body weight (kg)	78.8 ± 3.3	74.6 ± 10.0
BMI (kg/cm^2^)	26.0 ± 1.2	24.7 ± 2.6
Body fat percentage (%)	25.5 ± 3.5	23.8 ± 4.4
Systolic blood pressure (mmHg)	125 ± 5	124 ± 9
Diastolic blood pressure (mmHg)	85.0 ± 8.6	82.4 ± 10.8
Fasting blood glucose (mg/dL)	101.9 ± 10.2	98.8 ± 11.0
Fasting blood insulin (μU/mL)	4.29 ± 1.27	3.60 ± 1.55
HbA1c (%)	5.77 ± 0.14	5.61 ± 0.31
Triglyceride (mg/dL)	120 ± 74	98.5 ± 51.2
Total cholesterol (mg/dL)	217 ± 45	214 ± 35
LDL-cholesterol (mg/dL)	133 ± 42	129 ± 34

All data are presented as mean ± SD. BMI, body mass index.

**Table 5 foods-13-02628-t005:** Carbohydrate digestibility during in vitro digestion.

	C_∞_ (%)	k (min^−1^)	eGI
Placebo	93.8 ± 1.4 b	0.018 ± 0.003 a	105.6 ± 4.3 a
Treatment	102.5 ± 2.7 a	0.015 ± 0.003 a	107.9 ± 6.6 a

All data are presented as mean ± SD (*n* = 4). Different letters in the same column indicate significant differences (*p* < 0.05). C∞ = equilibrium starch hydrolysis percentage, k = kinetic constant, and eGI = estimated glycemic index.

## Data Availability

The original contributions presented in the study are included in the article/Appendix A, further inquiries can be directed to the corresponding author.

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
