# Peer review of "Impact of Rice Bran Oil Emulsified Formulation on Digestion and Glycemic Response to Japonica Rice: An In Vitro Test and a Clinical Trial in Adult Men"

_foods, 2024, doi:10.3390/foods13162628_

Round 1

Reviewer 1 Report

Comments and Suggestions for Authors

Very good and interesting study. I am wondering why the SD on the time course graphs in Figures 2,3 4 and 5are only showing on one side of the graphs. I believe they should show on both sides.

Author Response

Dear Reviewer 1

Thank you for your constructive advice.
Please confirm the attached documents for your review.

Reviewer 2 Report

Comments and Suggestions for Authors

Comments to the Authors

The manuscript “Impact of Rice Bran Oil Emulsified Formulation on Digestion and Glycemic Response to Japonica Rice: An In Vitro Test and a Clinical Trial with Venous Blood Sampling and Continuous Glucose Monitoring” is scientifically sound, and the results provided are well discussed highlighting also the limitations of this study and how to be faced.

However, some minor revisions need to be addressed.

Here are my detailed comments:

Abstract

-        Lines 13 and 18: “we performed”, “we conducted” I suggest to convert the first form in impersonal form. Please do it throughout all the manuscript.

Introduction

-        Lines 38: “exist”

-        Line 39: “is associated”

-        Line 46: Since BMI is reported for the first time in the text of the manuscript, add the meaning of the acronym.

-        The Introduction appears poor in state of the art. The novelty of this study needs to be better highlighted as well as how this study will contribute to the advancement of the knowledge on the topic when considering the researches already present in the literature. Moreover, the reasons why related to the choice of the rice bran oil is not well discussed.

Materials and Methods

-        Line 68: The refined rice bran oil? The sentence seems incomplete.

-        It is not clear how the optimal amount of refined rice bran oil has been selected. Please specify.

-        Line 123: Be coherent when reporting the number of participants. Line 99 it was reported in letters.

-        Paragraph 2.8.2: Which protocol have been followed? The standardized method INFOGEST static in vitro simulation of gastrointestinal food digestion (Brodkorb et al., 2019) has been considered?

Results and Discussion

-        Lines 210-212, equation (3), this part should be reported in the Materials and Methods section.

-        Line 282-283: “No significant differences were observed for all categories between the placebo and EMF test foods.” This statement is not supported by standard deviations and statistical analysis since not reported in Figure 7.

Conclusions

-        Line 371: substitute “suggest” with 2suggested” since the past tenses have been used.

-        Future perspectives should be reported and highlighted in the Conclusions section.

Author Response

Dear Reviewer 2

Thank you for your constructive advice.
Please confirm the attached documents for your review.

Reviewer 3 Report

Comments and Suggestions for Authors

After carefully reviewed this manuscript (foods-3123510), which involved a single-arm open clinical trial and in vitro testing to assess the effects of rice bran oil EMF on japonica rice in terms of digestion and glycemic response. Several concerns that should be addressed before proceeding further:

  1. The participant demographics in this study lack female representation, which is deemed unreasonable. Moreover, the selected age group of participants tends towards older individuals, predominantly middle-aged and elderly, with a noticeable absence of younger subjects (e.g., aged 20-30). This limitation restricts the generalizability of this study's findings, rendering them overly narrow in scope.
  2. Additionally, the experimental design of this study appears disorganized. For instance, stratified analysis mentioned in line 201 is not adequately detailed in the Methods section. Furthermore, the substantial error bars in your experimental results indicate significant individual variability among participants, thereby weakening the persuasiveness of your findings.

Other specific issues include:

  1. The title is overly lengthy and should be succinctly revised.
  2. In line 40, could you elaborate on what is meant by "social changes"?
  3. The introduction lacks sufficient detail on previous studies regarding the regulation of blood glucose by oil emulsified formulations. Citing only two references (ref. 11 and 12, lines 43-46) is insufficient.
  4. What is meant by "assess the eligibility of subjects" in line 53?
  5. Could you clarify the quantity of "a small amount of rice bran oil EMF" mentioned in line 60?
  6. I suggest either deleting or moving lines 84-95 to supplementary materials.
  7. How did you ensure that each participant "chewed 15 times per bite" as stated in line 106? This requirement seems difficult to control universally.
  8. Please explain the meaning of "−3 to 3" in line 120.
  9. The units for time (e.g., min and h) need to be standardized throughout the manuscript, as seen in lines 125-127.
  10. Could you elaborate on the significance of "2.7. Sample size" as mentioned in lines 131-135?
  11. Given that your study ultimately involved only 15 participants, there is no need to list "All participants (N = 29)" in Table 2. Furthermore, the absence of female participants in your study is considered unreasonable.
  12. The analysis of Table 5 is insufficient, and Figure 7 lacks error bars and significance analysis to support the conclusion "No significant differences were observed for all categories" (lines 282-283).

Comments on the Quality of English Language

Extensive editing of English language required.

Author Response

Dear Reviewer 3

Thank you for your constructive advice.
Please confirm the attached documents for your review.

Round 2

Reviewer 3 Report

Comments and Suggestions for Authors

I understand and acknowledge the author's explanation regarding the absence of female participants in the study. However, the author did not specify in the cover letter where the modifications were made in the manuscript (e.g., page number and line number), which makes it difficult for reviewers to accurately locate the changes and causes difficulty in reading. Nevertheless, the revised version is clearer than the previous one, and I recommend acceptance.

Comments on the Quality of English Language

 Moderate editing of English language required.